# New contributions of measurements in Europe to the global inventory of the stable isotopic composition of methane

Malika Menoud[1], Carina van der Veen[1], Dave Lowry[2], Julianne M. Fernandez[2], Semra Bakkaloglu[2,3], James L. France[2], Rebecca E. Fisher[2], Hossein Maazallahi[1,*], Mila Stanisavljević[4,*], Jarosław Nęcki[4,*], Katarina Vinkovic[5,*], Patryk Łakomiec[6,*], Janne Rinne[6,7,*], Piotr Korbeń[8,*], Martina Schmidt[8,*], Sara Defratyka[9,*], Camille Yver-Kwok[9,*], Truls Andersen[5,*], Huilin Chen[5,*], and Thomas Röckmann[1]

[1]Institute for Marine and Atmospheric research Utrecht (IMAU), Utrecht University, Utrecht, The Netherlands
[2]Greenhouse Gas Research Laboratory (GGRL), Royal Holloway, University of London, Egham TW20 0EX, United Kingdom
[3]Now at: Sustainable Gas Institute, Imperial College London, London SW7 1NA, United Kingdom
[4]Faculty of Physics and Applied Computer Science, AGH University of Science and Technology, Krakow, Poland
[5]Centre for Isotope Research, Energy and Sustainability Institute Groningen (ESRIG), University of Groningen, Groningen, Netherlands
[6]Department of Physical Geography and Ecosystem Science, Lund University, Sweden
[7]Natural Resources Institute Finland, Helsinki, Finland
[8]Institute of Environmental Physics, Heidelberg University, Heidelberg, Germany
[9]Laboratoire des Sciences du Climat et de l'Environnement (LSCE-IPSL) CEA-CNRS-UVSQ Université Paris Saclay, Gif-sur-Yvette, 91191, France
*These authors contributed equally to this work.

Correspondence: Malika Menoud (m.menoud@uu.nl)

**Abstract.** Recent climate change mitigation strategies rely on the reduction of methane ($CH_4$) emissions. Carbon and hydrogen isotope ratio ($\delta^{13}C_{CH_4}$ and $\delta^2H_{CH_4}$) measurements can be used to distinguish sources and thus to understand the $CH_4$ budget better. The $CH_4$ emission estimates by models are sensitive to the isotopic signatures assigned to each source category, so it is important to provide representative estimates of the different $CH_4$ source isotopic signatures worldwide.

We present new measurements of isotope signatures of various, mainly anthropogenic, $CH_4$ sources in Europe, which represent a substantial contribution to the global dataset of source isotopic measurements from the literature, especially for $\delta^2H_{CH_4}$. They improve the definition of $\delta^{13}C_{CH_4}$ from waste sources, and demonstrate the use of $\delta^2H_{CH_4}$ for fossil fuel source attribution. We combined our new measurements with the last published database of $CH_4$ isotopic signatures, as well as with additional literature, and present a new global database. We found that microbial sources are generally well characterised. The large vari-

ability in fossil fuel isotopic compositions requires particular care in the choice of weighting criteria for the calculation of a representative global value. The global dataset could be further improved by measurements from African, South American and Asian countries, as well as more measurements from pyrogenic sources.

We improved the source characterisation of $CH_4$ emissions using stable isotopes and associated uncertainty, to be used in top-down studies. We emphasise that an appropriate use of the database requires the analysis of specific parameters in relation

to source type and the region of interest.

The final version of the European $CH_4$ isotope database coupled with a global inventory of fossil and non-fossil $\delta^{13}C_{CH_4}$ and

$\delta^2 H_{CH_4}$ source signature measurements, is available at: (https://doi.org/10.24416/UU01-YP43IN (Menoud et al., 2022a).

## 1 Introduction

The current change of the Earth's climate is mainly caused by the emissions of greenhouse gases from anthropogenic activities (IPCC, 2013; IPCC 2021, 2021a). Methane ($CH_4$) is a strong greenhouse gas, with a global warming potential 32 times that of $CO_2$ over 100 years (Etminan et al., 2016). The increase in $CH_4$ concentration has contributed to an average warming of 0.5ºC in 2010-2019 compared to 1850-1900, which is slightly smaller than the contribution of $CO_2$ (IPCC 2021, 2021b). The global $CH_4$ mole fraction ($\chi(CH_4)$) in the atmosphere has drastically increased since 1984, when direct regular measurements

started, changing from 1645 ppb to 1850 ppb in 2017 (Nisbet et al., 2019). Compared to pre-industrial times (before 1750), the global $\chi(CH_4)$ has increased by 160%, from 720 to 1850 ppb (IPCC 2021, 2021a).

In the past 30 years, we have not observed a steady growth of atmospheric $CH_4$ mole fraction. Instead the increase in $\chi(CH_4)$ levelled-off between 2000 and 2007, and has been increasing again since then, from 2014 at the highest rate since the 1980's (Nisbet et al., 2019). This renewed increase presents a significant threat to reaching the goals of the Paris agreement, and miti-

gation policies are now also targeting $CH_4$ emissions (Shindell et al., 2017; Mayfield et al., 2017; Nisbet et al., 2020). Efficient strategies require good knowledge of the different kinds of $CH_4$ sources, their location and relative contributions. While emission estimates are reported at a country-level using statistical methods, atmospheric inversions, based on observations, can be used to verify the inventories (Houweling et al., 2000; Zavala-Araiza et al., 2015; Henne et al., 2016; Maasakkers et al., 2019). But the results from two approaches, respectively called bottom-up and top-down, are not in full agreement, reflecting a lack

in our understanding of the $CH_4$ cycle (Etiope and Schwietzke, 2019; Saunois et al., 2020; Stavert et al., 2021).

Measurements of $CH_4$ isotopologues provide additional constraints on the relative contribution of the various source categories, because $CH_4$ isotopic composition depends on the formation processes (Schoell, 1980; Whiticar, 1999; Quay et al., 1999). Time series of ambient $CH_4$ isotopic ratios are already used to derive emission scenarios in global models (e.g. Bousquet et al., 2006; Schaefer et al., 2016; Turner et al., 2017; Thompson et al., 2018; Fujita et al., 2020; Lan et al., 2021), and

at the regional scale (Röckmann et al., 2016; Stieger et al., 2019; Menoud et al., 2020, 2021; Varga et al., 2021). In addition, isotope measurements have proven to be very successful for source attribution in cities (Phillips et al., 2013; Zazzeri et al., 2017; Maazallahi et al., 2020; Xueref-Remy et al., 2020; Defratyka et al., 2021; Fernandez et al., 2022), and larger regions (Tarasova et al., 2006; Fisher et al., 2011; Beck et al., 2012; Warwick et al., 2016; Fisher et al., 2017; Lu et al., 2021). The uncertainties in the resulting emission rates of the different source categories depend on our knowledge of the different isotopic

source signatures, and understanding of their variability (Rigby et al., 2012; Schwietzke et al., 2016; McNorton et al., 2018; Szénási, 2020).

Direct measurements of the isotopic signature of $CH_4$ sources allow us to precisely characterise the type of emission, and a lot of data is available in the literature. Several review articles on $CH_4$ isotopic source signatures were previously published

(Rice and Claypool, 1981; Cicerone and Oremland, 1988; Bréas et al., 2001). The most recent one presented by Sherwood et al. (2017), and recently updated in Sherwood et al. (2021), gathered values from 13 489 locations (10 778 fossil fuel, 2711 non-fossil) from 347 published references. The 2017 study focused on (fugitive) fossil fuel sources, and allowed to re-evaluate the global $\delta^{13}C_{CH_4}$ value assigned to this emission category towards more depleted values (Schwietzke et al., 2016). A disadvantage of this database is that it is rather US-centered, and that the dataset is strongest for fossil fuel sources, but less robust for non-fossil sources. Therefore the database can be completed by more studies, especially concerning non-fossil sources.

The MEMO[2] project (MEthane goes MObile - MEasurements and MOdeling) was a H2020 MSCA European Training Network[1] with the goal to use innovative mobile measurement and modeling tools to improve the quantification of $CH_4$ emissions in Europe (Walter et al., 2019). An important component of MEMO[2] was the isotopic characterisation of $CH_4$ sources. Two laboratories involved in MEMO[2], at Utrecht University, The Netherlands, and at the Royal Holloway University of London, UK, carried out a large number of high-precision measurements with isotope ratio mass spectrometry (IRMS). Another method, using cavity ring-down spectroscopy (CRDS) was developed for the mobile measurements of ambient $CH_4$ isotopic composition. Several research groups were involved in field work with mobile measurements that targeted specific sources or environments in several European countries. Using this network, air samples from numerous $CH_4$ sources could be measured for isotopic composition. The resulting isotopic source signatures were gathered in a publicly available database, with the first version made accessible on October 1[st] 2020[2], and described in a publicly available report[3]. The European data was used in several publications over the past two years by Menoud et al. (2020, 2021, 2022b); Maazallahi et al. (2020); Defratyka et al. (2021); Bakkaloglu et al. (2021); Fernandez et al. (2022); Bakkaloglu et al. (2022). These studies emphasized the benefits from regional estimates of source $CH_4$ isotopic composition. The last update of the MEMO[2] isotopic data was compiled into The European Methane Isotope Database (EMID).

The present study provides an in-depth analysis of the EMID, a comparison with the global data, and the implications for the global understanding of $CH_4$ source isotopic composition. To this purpose, we compiled all the $CH_4$ isotopic source signatures from MEMO[2] with the latest version of the Sherwood et al. (2017, 2021) global database. We also searched the literature for more measured $CH_4$ source signatures to add to the dataset.

---

[1]Marie Skłodowska-Curie actions, Horizon 2020 Innovative Training Networks founded under the grant agreement No 722479: https://cordis.europa.eu/project/id/722479

[2]Menoud, M., Röckmann, T., Fernandez, J., Bakkaloglu, S., Lowry, D., Korben, P., Schmidt, M., Stanisavljevic, M., Necki, J., Defratyka, S., Kwok, C.Y., 2020. mamenoud/MEMO2_isotopes: v8.1 complete. Zenodo.

[3]Menoud, M., Röckmann, T., Lowry, D., Fernandez, J., 2020. Improved isotopic source signatures of local and regional $CH_4$ emissions (Deliverable No. 2.2), WP2. MEMO[2]: MEthane goes MObile – MEasurements and MOdelling, Available at: https://h2020-memo2.eu/wp-content/uploads/sites/198/2021/03/MEMO2-D2.2-v3-final.pdf.

## 2 Methods

### 2.1 Measurements within the MEMO$^2$ project

#### 2.1.1 Sampling

The data was collected by the research teams of eight universities and research institutes: Utrecht University (UU), the Royal Holloway University of London (RHUL), the Laboratoire des Sciences du Climat et de l'Environnement (LSCE), Heidelberg University (UHEI), AGH University of Science and Technology (AGH), Lund University (LU), the University of Groningen (UG), and the Netherlands Organisation for Applied Scientific Research (TNO). They participated in several campaigns in the Netherlands, the United Kingdom, France, Germany, Poland, Sweden, Romania and Turkey. Several other teams collaborated in two intensive campaigns: the CoMet[4] campaign in the Upper Silsian Coal Basin (USCB) in Poland (Fiehn et al., 2020; Gałkowski et al., 2021), and the ROMEO campaign in Romania[5] (Röckmann, 2020). The samples were collected mostly between 2017 and 2020, but three locations in the UK were sampled in February 2015, September and October 2016.

Different sampling methods were used:

- Mobile sampling on road vehicles, using a fast (0.1 to 10 Hz) analyser on-board to detect $CH_4$ enhancements (G2301, G2201-i, and G4302, Picarro Inc., USA; MGGA-918 and UGGA, Los Gatos Research, ABB, USA; LI-7810 Trace Gas analyser, LI-COR, USA; Dual Laser Trace Gas Monitor, Aerodyne Research, USA). Different setups were used by different teams with one or two of these instrument on-board, but the sampling procedure was the same. The samples were taken using a small electric pump connected to an inlet outside of the vehicle. The sample receptacles were bags of 1 to 3 L (Supel™-Inert Multi-Layer Foil bags, Sigma-Aldrich Co. LLC, USA; Tedlar or FlexFoil sample bags, SKC Inc., USA). Surveys were made around known sources of $CH_4$, where we sampled the elevated mole fractions as well as background $CH_4$ on the same day. If it was not practical to approach a source with the vehicle during mobile surveys, samples were taken on foot.

- Mobile sampling onboard of an aircraft, during the ROMEO campaign. A CRDS instrument (G4302, Picarro Inc., USA) was installed in the aircraft, and samples were taken from the outflow of the instrument into bags of 2 L (Supel™-Inert Multi-Layer Foil bags, Sigma-Aldrich Co. LLC, USA) when an increase in $CH_4$ mole fractions was observed. The method is described in detail in Menoud et al. (2022b).

- Mobile sampling on foot, without analyser. The samples were taken at regularly spread locations around a known $CH_4$ source, to make sure we collected air with $CH_4$ from the emission plume and background. In this case, the sample receptacles were bags of 2 to 3 L (Supel™-Inert Multi-Layer Foil bags, Sigma-Aldrich Co. LLC, USA; Tedlar sample bags, SKC Inc., USA), filled with a portable hand pump.

---

[4]Carbon dioxide and Methane mission, May-June 2018

[5]ROmanian Methane Emissions from Oil & gas, October 2019

– Soil chambers on wetlands in north Sweden and coal waste disposal areas in Poland. In wetlands, we installed transparent Plexiglas chambers on top of stainless steel collars that were pushed 20 cm into the peat. Samples from the chambers were taken during closure times, when $\chi(CH_4)$ increased, generally after 10 to 25 min. The soil chambers in Poland were made of plastic buckets covered with aluminum foil that were pushed about 5 cm in the ground and left for 30 min. In both cases, air was pumped into 2L sample bags (Supel™-Inert Multi-Layer Foil, Sigma-Aldrich Co. LLC, USA) for
further analysis in the lab.

– From an unmanned aerial vehicle (UAV), carrying an AirCore (coiled tubing) system to collect air samples (Andersen et al., 2018). The air samples were continuously pulled into the AirCore while flying transects across the plume of a $CH_4$ emission source, and were transferred to a 0.5 or 1 L bag sample after landing (Supel™-Inert Multi-Layer Foil, Sigma-Aldrich Co. LLC, USA) for further analysis in the laboratory.

**2.1.2 Measurements of isotopic composition**

The mass spectrometry measurements were performed at two laboratories: the IMAU (Institute for Marine and Atmospheric research Utrecht) at UU, and at the Department of Earth Sciences at RHUL. Both laboratories use a CF-IRMS (continuous flow isotopic ratio mass spectrometry) system to measure $\delta^{13}C$, and also $\delta^2H$ at IMAU. The system at IMAU was described by Röckmann et al. (2016) and the one at RHUL by Fisher et al. (2006). The reproducibility both groups can achieve is of 0.05
to 0.1 ‰ for $\delta^{13}C_{CH_4}$. At IMAU, $\delta^2H$ measurements have a reproducibility lower than 2 ‰. For consistency of the results, the two laboratories measured a set of 5 cylinders that contained air with $CH_4$ of different isotopic composition. The resulting differences in $\delta^{13}C_{CH_4}$ for each cylinder ranged between 0.02 and 0.04 ‰. They were within the analytical error reported by the two laboratories, so that the isotopic results obtained within the MEMO$^2$ project are consistent across the laboratories. The inter-comparison exercise is presented in detail in a MEMO$^2$ deliverable report, and publicly available [6].
The UHEI and LSCE groups performed isotopic measurements using CRDS instruments (G2201-i, Picarro inc., USA). Their measurement and calibration methods were described in Hoheisel et al. (2019) and Defratyka et al. (2021).

In the database, the method of isotopic measurements is specified by the "Measurement type" parameter, as either 'IRMS' or 'CRDS'. The laboratory where the measurements were performed is specified in the column "Measurement lab".

**2.1.3 Reported variables**

The analytical parameters reported in the database are $\delta^{13}C_{CH_4}$ and $\delta^2H_{CH_4}$, which are defined as:

$$\delta X = (\frac{R_{sample}}{R_{standard}} - 1)$$

---

[6]Lowry, D., Röckmann, T., Fisher, R., Menoud, M., Fernandez, J., 2018. Isotopic measurements linked to common scale (Deliverable No. 2.1), WP2. MEMO$^2$: MEthane goes MObile – MEasurements and MOdelling, Available at: https://h2020-memo2.eu/wp-content/uploads/sites/198/2018/12/MEMO2-D2.1-Isotopic-measurements-linked-to-common-scale-final.pdf.

with $R = \frac{^{13}C}{^{12}C}$ for $X = ^{13}$C or $R = \frac{^2H}{^1H}$ for $X = ^2$H

$\delta$ values are reported in per mille (‰), relative to the international standard materials Vienna Pee Dee Belemnite (VPDB) for $\delta^{13}$C, and Vienna Standard Mean Ocean Water (VSMOW) for $\delta^2$H.

### 135    2.1.4    Calculation of isotopic signatures

The measurement results of $\delta^{13}$C and $\delta^2$H of CH$_4$ are for ambient air, and not the sources themselves. There are different methods to derive the isotopic source signatures from the sampled CH$_4$ enhancement signatures; the Keeling plot and Miller-Tans methods are commonly used mass balance approaches. The Keeling plot method is based on the assumption that the background is stable during the sampling period (Keeling, 1961; Pataki et al., 2003). The Miller-Tans method is also applicable when the condition of a stable background is not fulfilled (Miller and Tans, 2003). Because background samples were taken on each survey day and in the same region, the condition of stable background was usually fulfilled. Defratyka (2021) showed that in this case, both methods lead to similar results within their uncertainty.

Both methods involve a linear regression model to fit the observed data. Different models were used: ordinary least squares (OLS) minimizing the difference in the y-axis coordinate, bivariate correlated errors and intrinsic scatter (BCES) (Akritas and Bershady, 1996), and orthogonal distance regression (ODR) (Boggs and Rogers, 1990). Zobitz et al. (2006) compared different regression methods to be applied in Keeling plots. The ODR method can induce a bias towards lower values, in the case the data points cover a relatively small range on the x-axis. Therefore, the OLS and BCES methods were usually preferred to calculate the source signatures for this study.

All the mass balance and regression methods are statistically valid. We did not work towards a uniform procedure, to not modify the data that was processed by each lab. The different approaches are specified for each entry of the database by the parameters "Mass balance approach" and "Regression method".

### 2.2    Revision of the global database of CH$_4$ isotope ratios

### 2.2.1    Structure of the database to include previous and new measurements

We used the same parameters as in the database of Sherwood et al. (2017, 2021) for non-fossil data. That is because our objectives concern only values for $\delta^{13}$C and $\delta^2$H of emitted CH$_4$, and do not include measurements of other gases or isotope signatures that Sherwood et al. (2017) reported in the fossil fuel database. The variables of interest are listed in Table 1 and include the site description (country, region, group, category and sub-category) and the $\delta^{13}$C and $\delta^2$H of CH$_4$. There are two types of values:

– Single measurement values, as from the characterisation of one emission event. Most fossil fuel data from Sherwood et al. (2021) are single measurements, as well as all the entries in the EMID.

– Average values from repeated measurements at the same location or over time. The values found in the literature are usually averages of multiple measurements.

A direct comparison between these two types of values would be unbalanced and lead to the over-representation of single measurements. Therefore, to combine the different kinds of data and perform statistical analyses, we aggregated the sources

165  reported in the EMID by region and sub-category, and in the fossil fuel database of Sherwood et al. (2021) per production basin. Throughout the article, the aggregated values are referred to as data *locations*, to distinguish them from *measurements* values which refer to the single events.

The source categories and sub-categories from Sherwood et al. (2017, 2021) were kept as they were, but when the new entries

170  from MEMO[2] measurements and published literature required it, we added additional source categories or sub-categories. The categories are grouped into the three main $CH_4$ formation pathways: modern microbial, pyrogenic, and fossil fuels. The "modern microbial" $CH_4$ is formed by microorganisms in surface ecosystems or in animals through enteric fermentation, and is referred to simply as "microbial" throughout the paper. Microbial $CH_4$ formation in the subsurface related to petroleum systems belongs to the "fossil fuels" category. Compared to Sherwood et al. (2017, 2021), we extended the "biomass burning"

175  category to "pyrogenic" to include emissions from other combustion sources, such as traffic or industry. All categories and sub-categories are listed in Table 2.

**Table 1.** Variables reported in the CH$_4$ isotopic signature database published with this article, which combines 3 datasets of different origins.

| Parameter | Description | Present in dataset | | |
|---|---|---|---|---|
| | | EMID | Sherwood et al. (2021), fossil fuel locations | Literature |
| CONTINENT | | x | x | x |
| COUNTRY | | x | x | x |
| STATE_REGION | administrative region or state | x | x | x |
| BASIN | Fossil fuel area | | x | |
| GROUP_TYPE | category level 3 | x | x | x |
| GROUP | category level 2 | x | x | x |
| CATEGORY | category level 1 | x | x | x |
| SUB-CATEGORY | category level 0 | x | x | x |
| SNAP | category in SNAP[1] | x | x | |
| LONG | longitude | x | | |
| LAT | latitiude | x | | |
| d13C_CH4_MEAN | $\delta^{13}C_{CH_4}$, in ‰ VPDB | x | x | x |
| d13C_CH4_ERR | error in the calculated $\delta^{13}C_{CH_4}$ | x | | |
| d13C_CH4_UNCERTAINTY | uncertainty in the reported $\delta^{13}C_{CH_4}$ | | | x |
| d13C_CH4_SD | standard deviation of $\delta^{13}C_{CH_4}$ | | x | |
| d13C_CH4_SE | standard error of the mean $\delta^{13}C_{CH_4}$ | | x | |
| d13C_CH4_N | number of $\delta^{13}C_{CH_4}$ values | x | x | x |
| d2H_CH4_MEAN | $\delta^{2}H_{CH_4}$, in ‰ VSMOW | x | x | x |
| d2H_CH4_ERR | error in the calculated $\delta^{2}H_{CH_4}$ | x | | |
| d2H_CH4_UNCERTAINTY | uncertainty in the reported $\delta^{2}H_{CH_4}$ | | | x |
| d2H_CH4_SD | standard deviation of $\delta^{2}H_{CH_4}$ | | x | |
| d2H_CH4_SE | standard error of the mean $\delta^{2}H_{CH_4}$ | | x | |
| d2H_CH4_N | number of $\delta^{2}H_{CH_4}$ values | x | x | x |
| TYPE_UNCERTAINTY | type of uncertainty reported | x | | x |
| COMMENTS | | x | x | x |
| REFERENCE | | x | x | x |

[1] SelectedNomenclatureforAirPollution,https://en.eustat.eus/documentos/elem_13173/definicion.html

**Table 2.** Number of measurements ($\delta^{13}C_{CH_4}$ / $\delta^2H_{CH_4}$) per source category in the updated $CH_4$ isotopic signature database.

| | | | | Sherwood et al. (2017) | Additional literature | MEMO[2] |
|---|---|---|---|---|---|---|
| microbial | agriculture | ruminants | C3/C4 | 227 / 86 | 45 / 12 | 30 / 11 |
| | | rice paddies | flooded, flooded seasonally | 360 / 139 | 15 / 0 | |
| | | piggery | | | 10 / 10 | |
| | waste | landfill | | 161 / 25 | 91 / 24 | 54 / 22 |
| | | sewage | wastewater, manhole | 2 / 2 | 27 / 6 | 83 / 64 |
| | | biogas | manure, C4/C3 | 15 / 15 | 21 / 2 | 39 / 8 |
| | | manure | cattle | | 9 / 0 | 22 / 0 |
| | | compost | | | | 4 / 0 |
| | | abattoir | cattle | | 18 / 9 | |
| | wetlands | temperate | marsh, bog, swamp, lake, estuary, pond, delta, fen, lagoon, reeds, flooded forest, wet prairie, river, mangrove | 246 / 124 | 150 / 8 | 6 / 6 |
| | | tropical | floodplain, lake, swamp, marsh, river, riverine reeds, mixed | 177 / 22 | 60 / 34 | |
| | | polar (incl. boreal) | bog, marsh, swamp, tundra, lake, estuary, fen, wet tundra, (thaw) permafrost, mire, forest | 558 / 14 | 72 / 2 | 15 / 15 |
| | other | termites | | 29 / 1 | 7 / 0 | |
| fossil fuels | exploitation | conventional | gas leak, gas installation, oil field, mixed, natural gas, oil refinery | 6517 / 2152 | 102 / 10 | 377 / 219 |
| | | coal | active coal mine, inactive coal mine, coal seam gas | 2108 / 796 | 113 / 71 | 71 / 40 |
| | | shale | | 447 / 290 | | |
| | seeps | oceans | marine seep | | | 4 / 4 |
| | | coal seam gas | | | 39 / 31 | |
| | | volcanoes | | | 0 / 8 | |
| pyrogenic | biomass burning | grass, pasture, brush, woodland, wood, forest, crop | C3/C4 | 109 / 4 | 1 / 1 | |
| | fossil fuel burning | conventional | car, traffic, residential heating | | 44 / 27 | 4 / 1 |

### 2.2.2 Data from previously published literature

We found an additional number of 48 sources[7] in the literature to complete the referred data listed in Sherwood et al. (2021). Because we aim at reflecting the actual $CH_4$ surface emissions to the atmosphere, we excluded studies that reported results from laboratory experiments, and of $CH_4$ dissolved in water (i.e. in oceans, wetlands and inland waters). We note that the search for data was biased because of the use of English language. The references we added concern published peer-reviewed articles and to a lesser extent thesis and conference papers. We did not perform additional data quality assessment. The studies were performed from 1982 to 2021 in various laboratories in the world. The study locations do not overlap with the ones of the EMID or the literature gathered in Sherwood et al. (2021), and we do not provide an analysis of potential temporal changes in the isotopic composition of the same source.

## 3 Results and discussion

The data on isotopic source signatures from the measurement campaigns carried out within the MEMO$^2$ project (2017-2020) were compiled into the EMID. The final version of this database is combined with the global database and additional literature, and is available at: https://doi.org/10.24416/UU01-4PO56T.

### 3.1 The European Methane Isotope Database (EMID)

The isotopic signatures obtained within the MEMO$^2$ project concern 734 locations over 8 countries, with $\delta^2H$ source signatures being measured at 54 % of the sites (Table 3). Measurements of $\delta^2H$ are less numerous than of $\delta^{13}C$ because only the measurement system at IMAU was able to measure this isotope signature. Depending on the availability of the measurement system, the sampling location and the timing of the campaign, it was not possible to systematically measure all samples at IMAU. Figure 1 shows the geographical distribution of the sampled sites in the different countries, according to the type of source. The number of sources we sampled does not represent the emission magnitudes.

During mobile surveys, we mostly targeted anthropogenic emissions from the exploitation and use of fossil fuels and waste processing facilities (Fig. 1). These are the most obvious anthropogenic $CH_4$ sources in densely populated regions, and we acknowledge a deliberate sampling bias towards urbanised areas. No biomass burning emissions were characterised during the MEMO$^2$ project. The EMID partially address the geographical bias pointed out by Sherwood et al. (2017): it particularly includes a large number of measurements made in Romania, where almost no data was available before.

---

[7]Kiyosu (1983); Chanton et al. (1989, 1992); Lansdown (1992); Wassmann et al. (1992); Gerard and Chanton (1993); Levin et al. (1993); Sugimoto and Wada (1993); Happell et al. (1994); Bergmaschi and Harris (1995); Happell et al. (1995); Chanton and Whiting (1996); Sugimoto et al. (1998); Bilek et al. (1999); Levin et al. (1999); Popp et al. (1999); Chanton et al. (2000); Chasar et al. (2000); Smith et al. (2000); Lowry et al. (2001); Chanton et al. (2002); Nakagawa et al. (2005); Bowes and Hornibrook (2006); Sugimoto and Fujita (2006); Hornibrook and Bowes (2007); Galand et al. (2010); Toyoda et al. (2011); Umezawa et al. (2011); Beck et al. (2012); Townsend-Small et al. (2012); Golding et al. (2013); Phillips et al. (2013); Baublys et al. (2015); Day et al. (2015); Iverach et al. (2015); Maher et al. (2015); Rella et al. (2015); Zazzeri et al. (2015); Owen et al. (2016); Zazzeri et al. (2016); Lopez et al. (2017); Obersky et al. (2018); Hoheisel et al. (2019); Lowry et al. (2020); Xueref-Remy et al. (2020); France et al. (2021); Lu et al. (2021); Al-Shalan et al. (2022)

**Table 3.** Number of $CH_4$ isotopic source signatures derived from sample measurements in the EMID.

|                 | $\delta^{13}C_{CH_4}$ | $\delta^2H_{CH_4}$ |
| --------------- | --------------------- | ------------------ |
| The Netherlands | 50                    | 27                 |
| United Kingdom  | 240                   | 54                 |
| Poland          | 98                    | 73                 |
| Germany         | 73                    | 23                 |
| France          | 46                    | 23                 |
| Sweden          | 21                    | 21                 |
| Romania         | 184                   | 174                |
| Turkey          | 2                     | 0                  |

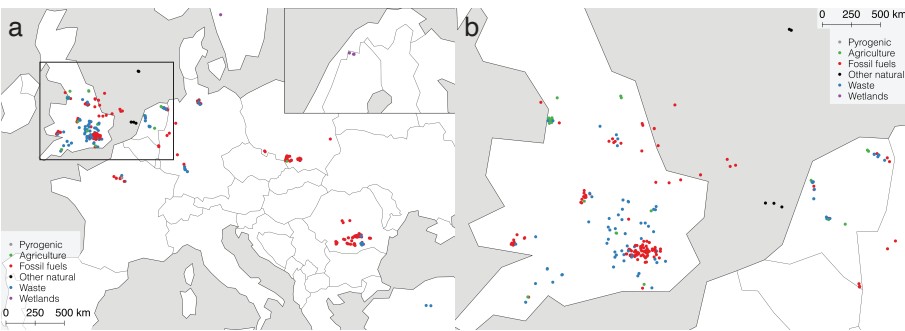

**Figure 1.** Geographical distribution of isotopic signature measurements ($\delta^{13}C$ and/or $\delta^2H$ of $CH_4$) carried out within the MEMO[2] project (2017 to 2020), depending on the type of source. **(a)** All locations. **(b)** Only in the UK, Netherlands and Germany.

We characterised 376 locations by both $\delta^{13}C$ and $\delta^2H$ values, and we compared the results to ranges reported in the literature in Fig. 2. The fossil fuel sources partly overlap with the range of thermogenic $CH_4$, but also spread towards lower $\delta^{13}C$ or higher
$\delta^2H$. This is due to the presence of natural gas of microbial origin in the coal reservoirs of Silesia, in Poland (Kotarba, 2001; Kotarba and Pluta, 2009; Menoud et al., 2021), as well as in Romania (Baciu et al., 2018; Fernandez et al., 2022; Menoud et al., 2022b). We concluded that this microbial $CH_4$ originates from the $CO_2$ reduction pathway as defined by Milkov and Etiope (2018), with relatively depleted $\delta^{13}C$ (<-60 ‰) and relatively enriched $\delta^2H$ (>-250 ‰). The $\delta^2H$ measurements were in these cases particularly useful to distinguish fossil fuels from microbial sources (Menoud et al., 2021; Fernandez et al., 2022;
Menoud et al., 2022b).

With an average $\delta^{13}C_{CH_4}$ of -53.6 ± 0.4 ‰ (n=202), the waste-related source signatures in the EMID generally have higher $\delta^{13}C$ values compared to typical microbial fermentation $CH_4$ (between -90 and -50 ‰; Milkov and Etiope (2018)). Waste sources measured in previous studies are less enriched, with an average of -56.0 ± 1.0 ‰ (n=56) in Sherwood et al. (2017). The average value in the EMID is strongly influenced by particularly enriched isotopic compositions in $CH_4$ emitted from
sewage water (range between -72.7 and -36.5, average of -50.5 ± 0.7, n=88) and to a smaller extent from biogas plants (range

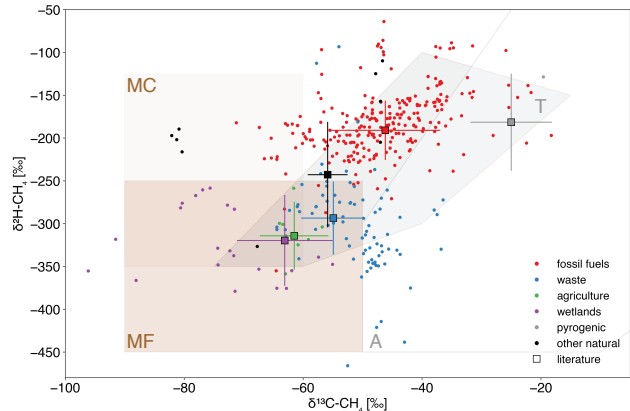

**Figure 2.** Dual isotope plot presenting measurement results from the EMID (circles); the literature data for the same source categories, taken from Sherwood et al. (2017, 2021) and completed with the mean and standard deviation values from additional publications (squares with error bars); shaded areas represent the different methanogenesis pathways from Milkov and Etiope (2018): MF = microbial fermentation, MC = microbial $CO_2$ reduction, T = thermogenic, A = abiotic.

between -64.4 and -45.5 ‰, n=54). A new study also reported surprisingly enriched $\delta^{13}C_{CH_4}$ (and $\delta^2H$) around a wastewater treatment plant in Australia: $\delta^{13}C$ = -47.6 $\pm$ 2 ‰ (Lu et al., 2021). Other recent studies in different regions of the world have also reported significantly higher $\delta^{13}C_{CH_4}$ from sewage plants compared to landfills (Hoheisel et al., 2019; Xueref-Remy et al., 2020; Al-Shalan et al., 2022). The $\delta^{13}C$ of $CH_4$ emitted from sewage treatment plants depends on process parameters:

oxic conditions lead to more enriched signatures than anaerobic treatment (Toyoda et al., 2011). Regarding biogas facilities, Bakkaloglu et al. (2022) emphasized the link between the type of substrate and the emitted $CH_4$ isotopic signatures: facilities that operate with C4 plant substrates emit $CH_4$ with higher $\delta^{13}C$ values in comparison with C3 plant substrates. Changes in waste management practices towards less disposal and more biogas production can likely explain the higher range of $\delta^{13}C$ values found in recent studies (Bakkaloglu et al., 2021). Another driver for more or less enriched $\delta^{13}C_{CH_4}$ emissions from waste

sources is isotopic fractionation when $CH_4$ reacts or diffuses. Diffusion and oxidation in the soil layers when $CH_4$ migrates from the deeper layers are secondary processes that cause isotopic fractionation (Bergamaschi et al., 1998; De Visscher, 2004; Conrad, 2005; Gebert and Streese-Kleeberg, 2017; Obersky et al., 2018; Bakkaloglu et al., 2021), which increases the range of possible isotopic signatures of the emitted $CH_4$.

The maps in Fig. 3 emphasize the similarities between $\delta^{13}C$ source signatures from modern microbial and fossil fuel sources in Poland and Romania. The average $\delta^{13}C_{CH_4}$ of fugitive emissions from fossil fuel extraction sites in Poland and Romania was -48.5 $\pm$ 0.6 ‰ (n=235), and of -55.3 $\pm$ 1.2 ‰ (n=42) for gas leaks and gas fields in Romania. From gas leaks in only the UK and the Netherlands, the average $\delta^{13}C_{CH_4}$ was -38.9 $\pm$ 0.3 ‰ (n=154), and -40.4 $\pm$ 0.3 ‰ (n=217) when including France and Germany, which reflect differences in the natural gas formation pathway compared to Poland and Romania. This

distinction is also visible in the histograms of the EMID in Fig. 5.a. In western Europe, $\delta^{13}C$ allows for a good separation

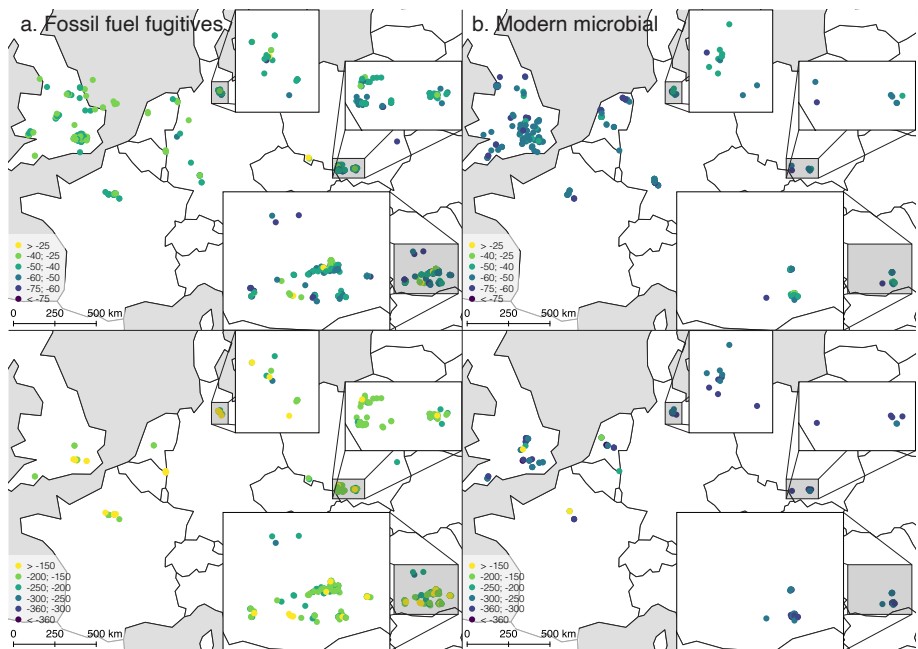

**Figure 3.** Measurement results of $\delta^{13}$C (top) and $\delta^2$H (bottom) in CH$_4$ from the EMID. **A.** CH$_4$ fugitive emissions from the exploitation of fossil fuels (gas leaks, oil and gas extraction and processing sites). **B.** CH$_4$ emissions from modern microbial fermentation sources (ruminants, landfills, sewage treatment plants and biogas plants).

between microbial and fossil fuel sources, which is well-established in the literature (Levin et al., 1993; Lowry et al., 2001; Röckmann et al., 2016; Zazzeri et al., 2017; Lowry et al., 2020). Yet we show that only $\delta^{13}$C data is not sufficient to distinguish microbial and fossil fuel CH$_4$ from all European regions. Fortunately, the $\delta^2$H$_{CH_4}$ source signatures allow for a clear distinction between fossil fuel and modern microbial emissions of anthropogenic origin (Fig. 3 and 5.a).

Previous isotopic measurements in Europe generally focused on western European countries (Levin et al., 1993; Bergamaschi et al., 1998; Lowry et al., 2001; Röckmann et al., 2016; Zazzeri et al., 2017; Cain et al., 2017; Fisher et al., 2017; Lowry et al., 2020; Xueref-Remy et al., 2020; Defratyka et al., 2021). This geographical bias should be addressed by focusing on western Balkan countries (Croatia, Bosnia, and Serbia) because of coal extraction activities (EDGAR inventory[8]), and densely populated areas in southern European countries such as Italy.

---

[8]European Commission, Joint Research Centre (EC-JRC)/Netherlands Environmental Assessment Agency (PBL), May 2021. Emissions Database for Global Atmospheric Research (EDGAR).

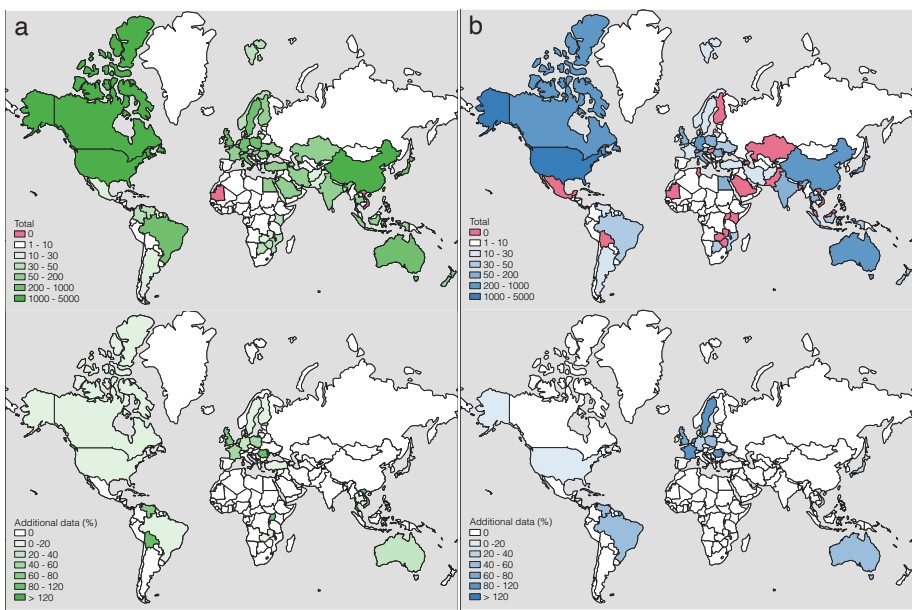

**Figure 4.** Number of isotopic signature measurements, **(a)** $\delta^{13}$C and **(b)** $\delta^2$H of CH$_4$, carried out in different countries. Top maps show the total numbers reported in the new global database. Bottom maps show the percentage of additional data brought by the EMID and the new published literature compared to Sherwood et al. (2017, 2021).

## 3.2 New global database

### 3.2.1 Overview and representativeness

The extended global database including all literature data and the aggregated MEMO$^2$ data consists of 13313 and 4337 measurements of $\delta^{13}$C and $\delta^2$H, respectively, from 64 countries. The map in Fig. 4 shows the partitioning of the measurement data per country, and Table 2 the number of records per CH$_4$ source. Table 4 contains statistics on the data from the EMID only, and the overall database including the EMID.

The number of measurements made in fossil fuel reservoirs and compiled in the database by Sherwood et al. (2021) is comparatively larger than from studies of other CH$_4$ emission sources (Table 1), and the number of measurements is not evenly spread geographically: significantly more measurements were made in North American and European countries, Australia, Brazil and Japan. In Russia and China, there were relatively more measurements as well, but only for fossil fuel sources. Despite including the first few measurements reported from Africa and the middle-east (France et al., 2021; Al-Shalan et al., 2022), the data distribution remains unbalanced. Nevertheless, specific isotope signatures dependencies can be further analysed for the different source categories:

**Fossil fuels** Fugitive emissions from fossil fuel reservoirs are highly variable not only on a large scale, but also from one basin to another, or even within the same basin (Sherwood et al., 2017; Milkov and Etiope, 2018; Alvarez et al., 2018;

Milkov et al., 2020a; Lan et al., 2021). Therefore, $CH_4$ isotopic composition from one basin can't be simply upscaled to a country scale. Any new isotopic measurement from a production basin with large fugitive $CH_4$ emissions brings relevant information. The recent measurements made in Romania, included in the EMID, illustrate well this heterogeneity (Menoud et al., 2022b).

    Sherwood et al. (2017) pointed out the lack of data for a list of conventional oil and gas and coal production countries,
in Africa, the middle-east, central and southern Asia, and South America. Previous estimates of global $CH_4$ isotopic signatures from the exploitation of fossil fuels weighted the source signatures from one basin by its fuel production (Schwietzke et al., 2016). Recent work suggest that fuel production is not a reliable proxy to estimate $CH_4$ fugitive emissions (Zavala-Araiza et al., 2015; Alvarez et al., 2018; Rutherford et al., 2021; Chen et al., 2021; Maazallahi et al., 2021). Thus, the most relevant sampling locations would be ideally related to estimated emission rates from top-down
measurements, instead of production or bottom-up emission estimates. Unfortunately, these data are lacking in many cases. Recently, particularly large $CH_4$ emissions were detected in central Asia (Varon et al., 2019), or measured in Mexico (Zavala-Araiza et al., 2021). Besides the new measurements in Romania, the EMID and additional literature we added to the global isotope database does not address the geographical representation issue.

**Modern microbial**   The isotopic signatures of $CH_4$ from modern microbial sources (mainly wetlands, ruminants, waste degra-
dation, rice paddies, termites) are largely dependent on environmental parameters such as the type of substrate and other ecosystem conditions. Figures A1 and A2 show that our new data confirm the trends previously observed: the $\delta^{13}C$ sensitivity to C3 or C4 plants in ruminant diet (Rust, 1981; Levin et al., 1993; Klevenhusen et al., 2010; Brownlow et al., 2017), to wetland latitudes ($\delta^{13}C$ depletion in polar regions because of less oxidation and the absence of C4 plants) (Fisher et al., 2017; Brownlow et al., 2017; Ganesan et al., 2018), and the $\delta^2H$ dependency on $\delta^2H_{H_2O}$ of precipitation,
and ultimately on the latitude (established for freshwater emissions) (Waldron et al., 1999; Chanton et al., 2006; Douglas et al., 2021; Stell et al., 2021). Based on the correlation with the plant metabolism (C3 or C4), $\delta^{13}C_{CH_4}$ from wetlands could be mapped on a global scale (Ganesan et al., 2018). Douglas et al. (2021) also suggested a spatial extrapolation of wetland $\delta^2H_{CH_4}$ using $\delta^2H_{H_2O}$ data, which can be interesting for under-sampled locations, for example in the southern hemisphere. However, a certain variability will always remain because of the influence of other parameters such
as the dominant methanogenic pathway (acetate fermentation or $CO_2$ reduction) (Waldron et al., 1998; De Visscher, 2004; Conrad, 2005; McCalley et al., 2014; Inglett et al., 2015; Chan et al., 2019; Douglas et al., 2021), or the $\delta^{13}C$ composition of the organic matter substrate (Conrad et al., 2011; Ganesan et al., 2018).

**Biomass burning**   Similarly to microbial degradation, the product of biomass burning is influenced by the plant constituents. $CH_4$ produced from the burning of C3 or C4 plants can be distinguished based on the $\delta^{13}C_{CH_4}$ values (e.g. Chanton
et al., 2000; Brownlow et al., 2017). Higher $\delta^{13}C$ signatures are measured when the burned plants are mostly C4 plants, and the $\delta^{13}C_{CH_4}$ is lower for C3 plants. This trend is clearly visible in the $CH_4$ isotope dataset, and is shown in Figure A3 of the supplementary material. The $\delta^2H_{CH_4}$ values are expected to depend to the $\delta^2H$ of local precipitations (Snover et al., 2000; Röckmann et al., 2010), but more measurements are needed to support this hypothesis.

**Table 4.** Statistical information on the results for the main $CH_4$ source categories of the EMID and the update of the global database including the EMID and additional literature data. "sd"=standard deviation, "se"=standard error of the mean.

| Variable | Statistic | Fossil fuel | | | | Modern microbial | | | | | Pyrogenic | |
|---|---|---|---|---|---|---|---|---|---|---|---|---|
| | | Conventional | Coal | Shale | All | Wetlands | Rice paddies | Ruminants | Waste | All | Biomass burning | Fuel combustion |
| EMID $\delta^{13}C$ | n events | 381 | 71 | | 457 | 21 | | 30 | 202 | 253 | | 4 |
| | mean | -43.8 | -48.7 | | -45.0 | -73.6 | | -63.0 | -53.6 | -56.4 | | -34.6 |
| | median | -42.0 | -48.9 | | -43.8 | -72.7 | | -62.9 | -53.3 | -55.6 | | -38.0 |
| | min | -71.2 | -65.4 | | -82.1 | -96.1 | | -73.9 | -72.7 | -96.1 | | -42.7 |
| | max | -19.6 | -18.3 | | -18.3 | -55.1 | | -56.8 | -36.5 | -36.5 | | -19.6 |
| | sd | 8.19 | 7.84 | | 8.93 | 10.4 | | 3.87 | 5.90 | 8.60 | | 10.3 |
| | se | 0.42 | 0.93 | | 0.42 | 2.27 | | 0.71 | 0.42 | 0.54 | | 5.15 |
| global $\delta^{13}C$ | n locations | 238 | 66 | 5 | 313 | 108 | 24 | 43 | 102 | 285 | 30 | 10 |
| | mean | -44.5 | -50.7 | -43.4 | -45.9 | -63.3 | -59.9 | -63.0 | -54.6 | -59.8 | -26.1 | -22.7 |
| | median | -42.9 | -50.9 | -42.2 | -44.6 | -63.1 | -59.5 | -63.3 | -54.3 | -59.0 | -27.2 | -20.3 |
| | min | -77.4 | -72.9 | -49.3 | -77.4 | -88.9 | -67.2 | -74.4 | -73.9 | -88.9 | -33.4 | -39.6 |
| | max | -18.9 | -25.6 | -39.5 | -18.9 | -44.4 | -50.8 | -50.3 | -45.1 | -44.4 | -12.5 | -9.00 |
| | sd | 8.44 | 10.4 | 3.84 | 9.16 | 8.17 | 4.53 | 5.31 | 4.90 | 7.61 | 5.24 | 11.2 |
| | se | 0.55 | 1.28 | 1.72 | 0.52 | 0.79 | 0.92 | 0.81 | 0.49 | 0.45 | 0.96 | 3.55 |
| EMID $\delta^2H$ | n events | 220 | 40 | | 268 | 21 | | 11 | 94 | 126 | | 1 |
| | mean | -181 | -185 | | -182 | -325 | | -310 | -305 | -309 | | -129 |
| | median | -184 | -184 | | -185 | -337 | | -304 | -303 | -307 | | -129 |
| | min | -355 | -271 | | -355 | -379 | | -359 | -466 | -466 | | -129 |
| | max | -85.8 | -63.8 | | -63.8 | -258 | | -259 | -93.2 | -93.2 | | -129 |
| | sd | 39.5 | 30.7 | | 39.1 | 41.2 | | 25.6 | 54.8 | 51.1 | | |
| | se | 2.7 | 4.9 | | 2.4 | 9.0 | | 7.7 | 5.7 | 4.6 | | |
| global $\delta^2H$ | n locations | 118 | 37 | 4 | 164 | 32 | 4 | 13 | 41 | 92 | 5 | 6 |
| | mean | -183 | -210 | -147 | -189 | -319 | -323 | -310 | -292 | -306 | -226 | -136 |
| | median | -179 | -208 | -140 | -187 | -309 | -328 | -308 | -301 | -308 | -210 | -126 |
| | min | -263 | -310 | -191 | -349 | -472 | -336 | -404 | -344 | -472 | -285 | -192 |
| | max | -101 | -162 | -116 | -101 | -246 | -301 | -224 | -113 | -113 | -195 | -81.0 |
| | sd | 32.4 | 27.4 | 32.1 | 35.8 | 53.2 | 15.6 | 45.0 | 45.7 | 48.3 | 35.8 | 39.4 |
| | se | 3.0 | 4.5 | 16.0 | 2.8 | 9.4 | 7.8 | 12.5 | 7.1 | 5.0 | 16.0 | 16.1 |

### 3.2.2 Global data and the EMID

Statistical information on the $CH_4$ isotopic signatures in the complete extended database are presented in Table 4. Fig. 5 shows the distribution frequency of isotope signatures for the source categories that represent the largest reported emissions (Saunois et al., 2020). The categories agriculture, waste, wetlands, and partly other natural are all of modern microbial origin, mostly

from acetate fermentation (Milkov and Etiope, 2018). The different categories within microbial processes generally overlap (Figure 5). Some differences can however be observed, such as the wetlands mean $\delta^{13}C_{CH_4}$ being lower in the EMID than globally (-73.6 $\pm$ 2.27 ‰ compared to -63.3 $\pm$ 0.79 ‰), because the European samples were taken at relatively high latitudes (section 3.2.1). Table 4 also shows that waste sources present more enriched $\delta^{13}C_{CH_4}$ values than other modern microbial sources. This diffference is particularly visible in the EMID, where a relatively large number of sites from waste related sources were sampled. As mentioned in section 3.1, additional parameters control the isotopic signature of the emitted $CH_4$, such as the type of substrate, the presence of oxygen, or secondary (e.g. oxidation) processes. The minimum waste $\delta^{13}C$ signature of -73.9 ‰ is comparable to the low values of other microbial sources, which supports the hypothesis of a larger influence of secondary processes in waste degradation relative to other microbial $CH_4$ formation. We recommend to separate the waste category from the other microbial sources to minimise the uncertainty in the assigned isotopic signature, at least for $\delta^{13}C$. The range of $\delta^2H$ signatures from waste sources is larger than of the other modern microbial sources, but the average $\delta^2H$ from the different microbial sources are similar. One can see that $\delta^2H$ is not systematically correlated with $\delta^{13}C$, and $\delta^2H$ can also vary with other parameters such as the isotopic composition of water in the substrate. The $\delta^2H$ signatures for waste are based on less measurements compared to $\delta^{13}C$ (42 % of all measured waste sources included $\delta^2H$ signatures). The relation between $\delta^2H_{CH_4}$ from wetlands and the $\delta^2H_{H_2O}$ from precipitation has been established previously (Waldron et al., 1999; Chanton et al., 2006; Douglas et al., 2021). We also know that the fractionation factors derived for $CH_4$ microbial oxidation are much larger for $\delta^2H$ than for $\delta^{13}C$ (Coleman et al., 1981; Bergamaschi et al., 1998; Chanton et al., 2006). Nevertheless, further $\delta^2H$ measurements are required to better define the isotopic dependancies to secondary processes.

In Sherwood et al. (2017, 2021), the pyrogenic category only contained biomass burning data, and the binary distribution clearly illustrates the difference between C3 and C4 plants in terms of $\delta^{13}C_{CH_4}$ signatures: the averages in the global database are -28.4 $\pm$ 0.65 and -18.0 $\pm$ 1.9 ‰, for C3 and C4 plants, respectively. The additional biomass burning data we added from published literature confirms the dependency of $\delta^{13}C_{CH_4}$ on the plant metabolism (Figure A3). We also added pyrogenic data from fuel combustion (burning of fossil fuel) from both our measurements and the literature. The resulting distribution of the $\delta^{13}C$ data is smoother than in Sherwood et al. (2017) (Fig. 5), because the $\delta^{13}C_{CH_4}$ from fossil fuel burning does not show a clear distinction between C3/C4 plant metabolisms. $\delta^2H_{CH_4}$ isotopic signatures from pyrogenic sources cover a wide range of values, and overlap with the ones of fossil fuels. $\delta^2H$ signatures allow to clearly distinguish between biomass and fuel combustion (Table 4), but this is based on a very low number of measurements. Further analysis including data on $\delta^2H_{H_2O}$ could help to parametrise the biomass burning $\delta^2H_{CH_4}$ in more detail (Vigano et al., 2010), similar to the above mentioned relation between $\delta^2H_{CH_4}$ and $\delta^2H_{H_2O}$ (Waldron et al., 1999; Chanton et al., 2006; Röckmann et al., 2010; Douglas et al., 2021).

Fugitive $CH_4$ emissions from fossil fuel source locations present a wide range of isotopic signatures: $\delta^{13}C$ from -77.4 to -18.9 ‰ and $\delta^2H$ from -349 to -101 ‰ (Table 4). The average signatures of all fugitive $CH_4$ emissions from the exploitation of fossil fuels (excluding seeps) in the EMID were $\delta^{13}C$ = -44.6 $\pm$ 0.4 ‰ (n=452) and $\delta^2H$ = -182 $\pm$ 2.4 ‰ (n=259), which compares well with the global average of -44.8 $\pm$ 0.1 ‰ (n=8128) calculated in Sherwood et al. (2017). Regarding the present

**Table 5.** $CH_4$ isotopic source signatures assigned to the fossil fuel related emissions in past global scale models (not exhaustive list)

| Reference | $\delta^{13}$C VPDB [‰] | $\delta^2$H VSMOW [‰] |
|---|---|---|
| Gupta et al. (1996); Tyler et al. (2007) | -38 / -37[1] | -175 |
| Neef et al. (2010); Monteil et al. (2011) | -40 / -35[1] | |
| Rigby et al. (2012) | -40[2] | -175 |
| Rice et al. (2016) | -41.7 | -175 |
| Schaefer et al. (2016) | -37 | |
| Schwietzke et al. (2016) | -44 | |
| McNorton et al. (2018) | -42.6 | |
| Fujita et al. (2020) | -45.2 | -209 |
| This database, mean $\pm$ sem[3] | -44.5 $\pm$ 0.5 / -50.7 $\pm$ 1.3[1] | -182 $\pm$ 3.0 / -210 $\pm$ 4.5[1] |

[1] for natural gas/coal; [2] also in Lassey et al. (2000); Houweling et al. (2000); Bousquet et al. (2006); Gosh et al. (2015); Thompson et al. (2018); [3] standard error of the mean.

updated global database, the weighted averages were $\delta^{13}$C = -46.6 $\pm$ 1.8 ‰ and $\delta^2$H = -192 $\pm$ 7.5 ‰, weighted by the relative emissions from conventional and coal fuels production worldwide[9]. The average values from the different databases are lower than $\delta^{13}$C and $\delta^2$H values used in most global models (Table 5), and to the value of -44.0 $\pm$ 0.7 ‰ suggested by Schwietzke et al. (2016). The global means in Table 4 do not necessarily represent the global isotopic signature of fossil fuel emissions, because this should be weighted by the magnitude of emissions in the different basins, which was taken into account (using production as indicator) in the calculation by Schwietzke et al. (2016). However, our averages are indications of the general $CH_4$ isotopic signatures from all measurements until now. Because of the high heterogeneity of the $\delta^{13}C_{CH_4}$ and $\delta^2H_{CH_4}$ values from fossil fuel related activities, and the temporal variations in the production from the different regions (Stavert et al., 2021; US Energy Information Administration, 2021; Lan et al., 2021), it is important to assume a relatively large uncertainty when estimating in the global signature of fossil fuel emissions in atmospheric models.

In section 3.1, we have shown the use of $\delta^{13}C_{CH_4}$ to distinguish fossil fuel emissions in western Europe, and the need for $\delta^2H_{CH_4}$ measurements in central and eastern Europe. In the global database, most fossil fuels records (83.5%) have $\delta^2H_{CH_4}$ values >-250 ‰. The few values of $\delta^2$H <-300 ‰, indicating microbial fermentation as gas origin, were found in some coal formations in the United States and Canada. Figure 5 still allows us to generally conclude that $\delta^2$H measurements are more suitable to distinguish fossil fuel vs. biogenic $CH_4$ sources at the global scale than $\delta^{13}$C only, which further emphasizes the need for more $\delta^2H_{CH_4}$ measurements.

The extraction of shale gas is growing worldwide (Energy Information Administration, 2016, 2021), as well as the associated $CH_4$ emissions (Howarth, 2019; Milkov et al., 2020b). However, shale gas commercial production does not increase in Europe (Energy Information Administration, 2016), and so the emphasis of this study is limited to oil, gas and coal fuels.

[9]Relative weights of 0.66 for conventional fuels (oil and natural gas) and 0.34 for coal. Emission data from Saunois et al. (2020)

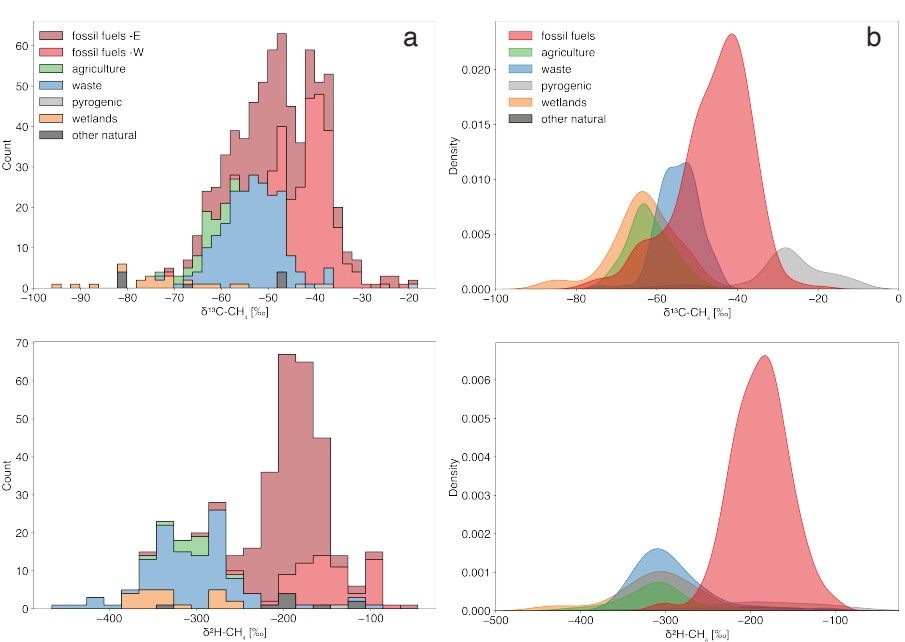

**Figure 5.** Distribution of $\delta^{13}$C (top) and $\delta^2$H (bottom) in $CH_4$ for different source categories. **(a)** Single measurements reported in the EMID (absolute numbers). "fossil fuels -E" shows fossil fuels data from Poland and Romania, and "fossil fuels -W" from the UK, the Netherlands, Germany, and France. **(b)** measured locations in all datasets (Sherwood et al., 2021), with EMID locations and additional literature) (normalised probability density). "agriculture" represents ruminants and rice paddies emissions.

## 4 Conclusions

This study presents an updated dataset of isotopic source signatures of $CH_4$ from recent atmospheric measurements, while including additional data from published literature which were not previously included. The new data is a contribution from the EMID, that results from the sampling activities performed within the MEMO$^2$ project. It represents a substantial contribution to the global dataset for fugitive fossil fuels and waste sources, mainly sampled in urban areas.

We have highlighted two main improvements in our understanding of the $CH_4$ isotopic composition: (i) A more robust range of values for modern microbial sources, and a better characterization of the $\delta^{13}C$ enrichment in $CH_4$ from waste sources. (ii) Fossil fuel related sources could have more depleted values than previous estimates used in global models. In this respect, our data confirm the analysis made by Schwietzke et al. (2016).

Finally, the new European data contain comparatively more $\delta^2H$ measurements. In the case of fossil fuel emissions, the use of $\delta^2H_{CH_4}$ is of particular interest. In general, utilizing both $\delta^{13}C$ and $\delta^2H$ for $CH_4$ improves our ability to clearly separate fossil fuel and microbial sources, compared to $\delta^{13}C$ alone. The use of $\delta^2H$ as additional constraint could help to answer open questions regarding the $CH_4$ global budget. To better understand the drivers of $\delta^2H$ variability (except for $\delta^2H$ of precipitation), more measurements are required, especially of pyrogenic and waste sources.

The present dataset can be used for $CH_4$ source attribution, studies at local and regional scales, and to derive global source signatures for input to global methane cycle modeling studies. The larger dataset will also help to estimate the uncertainties to take into account when using isotopic data in top-down studies, and with prior knowledge of the specificities of the studied region, the use of isotopic data in top-down studies is a powerful tool to evaluate the bottom-up emission inventories (Alvarez et al., 2018; Etiope and Schwietzke, 2019; Rutherford et al., 2021; Stavert et al., 2021). A future improvement of this database would be to include more measurements on the African, Asian and South American continents, where experimental studies are lacking. Because of its potential for source characterization, new studies should also focus on $\delta^2H_{CH_4}$ measurements. The maintenance of a $CH_4$ stable isotope database relies on a certain transparency of different groups around the world on their work. Therefore we strongly encourage the scientific community to pursue the efforts to make scientific data open access more systematically.

## 5 Data availability

The database is made freely available to the scientific community in the belief that it provides the most complete picture of the stable isotopic composition of $CH_4$ sources. The free availability of these data does not constitute permission for publication of the data. For research projects, if the data used are essential to the work to be published, or if the conclusion or results largely depend on the data, co-authorship should be considered. Full contact details and information on how to cite the data are given in the accompanying database. The database is currently stored in a publicly available repository: https://doi.org/10.24416/UU01-YP43IN (Menoud et al., 2022a).

## Appendix A: Supplementary material

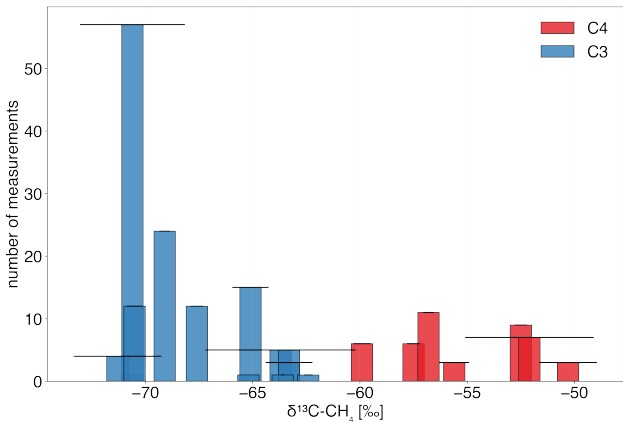

**Figure A1.** Measured $\delta^{13}C_{CH_4}$ signatures from ruminants in the literature studies[1] according to the feed: a majority of C3 plants (blue) or C4 plants (red). Bar heights represent the number of measurements and black lines standard deviations.

[1] Al-Shalan et al. (2022); Brownlow et al. (2017); Klevenhusen et al. (2009, 2010); Levin et al. (1993); Lu et al. (2021); Rust (1981); Townsend-Small et al. (2012); Wahlen et al. (1989)

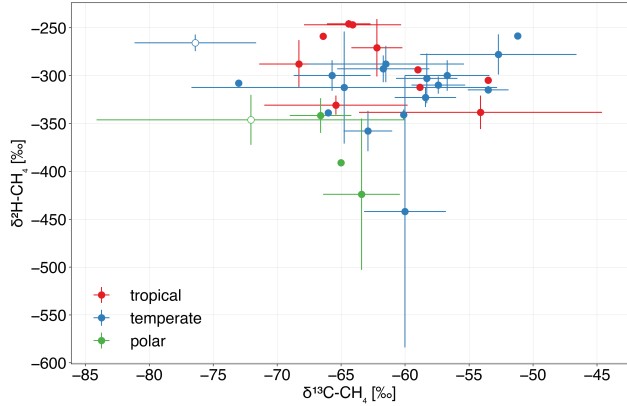

**Figure A2.** Measured $\delta^{13}C_{CH_4}$ and $\delta^2H_{CH_4}$ signatures from wetlands sites as reported in the literature[2] (solid circled) and EMID (open circles) database, color coded by the latitude zones. Error bars show the standard deviations.

[2] Beck et al. (2012); Burke and Sackett (1986); Day et al. (2015); Happell et al. (1994); Kuhlmann et al. (1998); Lansdown (1992); Levin et al. (1993); Martens et al. (1992); Nakagawa et al. (2002); Smith et al. (2000); Sugimoto and Fujita (2006); Umezawa et al. (2011); Wahlen et al. (1989); Wassmann et al. (1992); Woltemate et al. (1984)

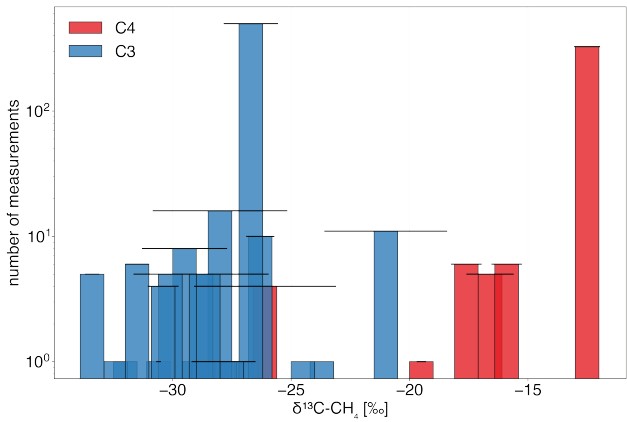

**Figure A3.** Measured $\delta^{13}$C-CH$_4$ signatures from biomass burning in literature studies[3] according to the type of vegetation: a majority of C3 plants (blue) or C4 plants (red). Bar heights represent the number of measurements and black lines standard deviations.

[3] Stevens and Engelkemeir (1988); Wahlen et al. (1989); Levin et al. (1993); Chanton et al. (2000); Snover et al. (2000); Fisher et al. (2011); Umezawa et al. (2011); Brownlow et al. (2017)

*Author contributions.* MM, CV, DL, JF, SB, JF and RF performed the isotopic measurements. MM, TR, DL, JF, SR, JF, RF, HM, MS, JN, KV, PŁ, PK, MS, SD and TA took part in the collection of samples. MM gathered and analysed the data and prepared the figures; TR and DL contributed to the interpretion of the data. MM prepared the manuscript with contributions from TR, DL, JR, MS, PL, SB, HM, JF and HC.

*Competing interests.* The authors declare that they have no conflict of interest.

*Acknowledgements.* We thank all the staff from different organisations involved in the MEMO$^2$, CoMet and ROMEO projects who participated in the sample collection. We acknowledge the technical staff at UU and RHUL for the maintenance of the IRMS systems.

This work was supported by ITN project "Methane goes Mobile – Measurements and Modelling" (MEMO2; https://h2020-memo2.eu/, last access: November 3$^{rd}$, 2021). This project has received funding from the European Union's Horizon 2020 Research and Innovation programme under the Marie Sklodowska-Curie grant agreement no. 722479.

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
