# Peer review of "New contributions of measurements in Europe to the global inventory of the stable isotopic composition of methane"

_Earth System Science Data, 2022_

## Referee Comment (RC2)

Review on Menoud et al., essd-2022-30.

**General comments**
The article presents the data of methane isotopic air samples collected in Europe, their analysis, and discussions in respect to previous studies of global inventories/data. The isotopic information for various sources in methane are useful in understanding the methane budgets better. Atmospheric inverse models (top-down methods) would especially benefit from such information – not only closing the total budgets based on total methane concentration data, but also allow to separately estimate emission magnitude of different source sectors. As authors point out, the previous studies on methane isotope measurements were mainly based on samples from US. The additional data from Europe therefore increases geographical representation, and is highly valuable. The paper presents important information and data for carbon cycle community, and is worth publishing. I also appreciate the authors for making the data open access.

I have a few suggestions below to improve the presentation of the manuscript that would possibly increase the value of the paper.

- Please consider rephrasing the title. To me, updating global inventory is a by-product, and the European data collected/presented here is the most important part.
- Please consider focusing more on Europe. You could, for example, add information and discussion about European data from previous studies, i.e. validation and updating information on Europe. It is unclear from the current manuscript how much information/data were available in Europe previously, and what were the isotopic values. You can also add comparison of the European data to global data, i.e. discussion on geographical differences/similarities of isotopic composition. In addition, geographical bias still remains within Europe. Please comment on it – what should we still improve in Europe? Where would be critical locations, and from which sources that we should sample data from, and why?
- Please consider focusing on new information brought by newly sampled and collected data. A few suggestions/comments on this are found under Specific comments belo.
- Please add temporal information about the collected data in MEMO$^2$ and those compared (collected from previous studies). If those are from very different period, the differences in isotopic composition may indicate temporal changes in underlying processes. Please comment on such if any.
- Modern microbial section: in many cases, $CH_4$ emissions from natural sources (e.g. peatlands) are separately estimated from anthropogenic sources (e.g. waste, agriculture). It would be more helpful if you could analyse those separately. Could you separate those e.g. when comparing to previous studies (incl. Section 2.5)?

**Specific comments**
Please check language in general. Sometimes informal structures/phrases are used.

Please check the journal criteria for citations. There are a few "in review" papers that are cited several times, but the manuscript/preprint are not available. Note that those should not be included in the final version of this manuscript. Please acknowledge that some comments maybe senseless because of this, and I apologies for that. In addition, some are cited within a page, and some at the end of the manuscript. I think all should be given at the end of the manuscript.

P1 L2: "measurements" → "isotopic measurements"

P2 L44: What do you refer to by "them"?

L3 L60-61: "numerous $CH_4$ sources could be sampled for isotopic measurements"
I think what you sample is air, and not the sources. Please revise the sentence.

Method:
In the excel file, I see that in some locations, only $^{13}C$ or H isotopes are sampled/analysed (for MEMO$^2$ data). Why both were not sampled at all locations? Could you add information about selection criteria or sort if there was such?

Section 2.1.1
- Please consider adding a table about different sampling methods, and give focus to the differences/similarities in those methods in the text. This way, you could reduce amount of details and avoid duplicates in the text.
- Was there any location where different sampling methods were applied at the same place? What kind of differences would occur in the measured values due to the differences in sampling methods?

P5 L116-119:
I am now confused. In the beginning of this section (2.1.2), it is written that "the mass spectrometry measurements were performed at **two** laboratories". However, is it so that actually four labs analysed the air samples? You have included detailed comparison and measurement precision on IRMS, but how about CRDS compared to IRMS? Was there any differences between UHEI and LSCE measurements?

P5 L133: "we did not work towards a uniform procedure"
Why? Did you allow e.g. each lab to calculate based on their choice of method? All methods being "statistically **valid**" does not mean that there are no differences. So I do not see why this is a valid statement to use different procedures.

P6 L139: "our objective concern only values … of emitted $CH_4$"
Do you mean that the forests, where samples are taken from, also emit $CH_4$? Most of forests are net sink of $CH_4$, but in some cases they emit $CH_4$. Sorry for this picky comment, but I am simply interested in. Could you provide the references where those samples were taken from?

P6 L141-143:
I do not see country/region in Table 1. What is the "region" in those sentences?

P8 L164: (2017-2020)
Is this the project period or actual period when data was collected? As in the earlier comments, it would be informative to state clearly the sampling period.

P8 L165-167: "The first version was made accesssible on October 1st 2020"
Now that it is updated, is this information needed?

P8 L166: "The European data was used in several publications"
This gives an impression that the data has been published already. Do you mean the first version of the data was used in those publications? If so, why did you need to update? Please also consider moving this to Introduction by add a bit more details about those studies, e.g. how and what the data (e.g. which isotopologue/country) were used, and what are the main findings. This would then show the importance of the European dataset and what new information it can bring.

P9 L183-187:
What were then the isotopic signature values for those coal reservoirs where natural gas of microbial origin are present? You mention "a relatively enriched $\delta^2H$ (>-250 ‰), and relatively depleted $\delta^{13}C$ (<-60 ‰)", but are those values those measured in this study or from Milkov and Etiope (2018)?

P10 L190-202:
Similar to the comment above. What is the value found in this dataset? You mention that "microbial fermentation range specified in previous reviews", but how were the waste-related source signatures compared to fermentation range measured in this study? What are the mean and rage of signature values in sewage treatment plants and biogas plants? How are those values compared to those in other types of plants?

P10 L207-P11 L208: "This distinction is also visible in the histograms of the European Methane Isotope Database in Fig. 5.A"
- "Fig. 5.A." → Fig. 5(a)"
- From the figure, those from Poland and Romania have two peaks. Does this mean that within Poland and Romania, there are different types of plants, one similar to those in the UK/Netherlands and another that is microbial origin?

P10 L208-209: "In western Europe, $\delta^{13}C$ allows for a good separation between microbial and fossil fuel sources"
Is it true? I see a quite much overlap still.

P11 L216-217: "increase in number of measurements"
I do not see from the figure how much data is increased, but only the total. Is the Figure caption or text wrong?

Section 2.5:
Much of the text seems to be more suitable as Introduction. It is unclear what is the results of this study in addition to Sherwood et al (2021). Did representativeness increased by additional data found in this study?

Section 2.6:
There are a lot of discussion where newly sampled and collected data are compared to Sherwood et al. (2017, 2021) values. However, I feel that it is difficult to comprehend the differences from the current figures and tables. Could you consider adding figures corresponding to e.g. Fig. 4 and 5, but showing e.g. differences to the previous study? Such figures could be in the supplementary, but would be helpful for readers who does not remember all the details in Sherwood et al. (2017, 2021).

P13 L255: "mainly following the fermentation pathway"
Fermentation pathway applies to agricultural sources, but not for e.g. wetlands. Please revise the sentence.

P13 L256: "They show a normal distribution"
Have you checked whether they really are normally distributed?

P13 L256-258:
I see quite much overlap in waste sector to, e.g. agriculture, also in MEMO$^2$ data.

P13 L271-273:
I am not sure what you wish to emphasise here. Why $\delta^{13}$C-CH$_4$ from fossil fuel burning being "more variable than biomass burning" lead to "smoother" distribution?

P13 L273-276 (and P13 L264-267):
I feel this relation and need of additional data on $\delta^2$H-H$_2$O is more suitable to be mentioned in Conclusions.

P14 L280: "weighted average"
What is the weight used? If it is emission weighted, did you also measure emissions at the same locations?

P15 L297-301:
This is more suitable to be mentioned in Conclusions.

Conclusions: Now that the MEMO$^2$ project has ended (if I understood correctly), is there a plan to continue activities on isotopic measurements?

**Figures and tables**
Figure 2: The literature values seem to be illustrated by boxplot-type. Are those squares mean/median? What does the bar length illustrate? The letters presenting the types of shaded areas (e.g MF) can be bigger, and the letter "T" is better to be straight. I cannot see clearly the area "A", but is it a white space on the right side?

Table 3:
   • Please add/separate global to European means.
   • What is "sem"?

Please consider adding A2 in the main text.

**Technical comments**
P2 L17: "the earth's" → "the Earth's"

P5 L115: "deliverable report publically availabe" → "deliverable report, and publically available"

---

## Author Response (AR1)

**Reviewer 1: General comments**

An important number of new measurements are reported to help inform a global methane isotope ratio database. These measurements were made as part of a European project on methane, accruing a large number of measurements over the project's period. Interesting and significant findings are explained regarding the variability of isotope ratios over the region and within source categories. These findings are then put in the context of the global measurements made and reported across the literature.

The continuing improvement of our understanding of isotopic source signatures for methane isotope ratios is vital if atmospheric measurements are to make accurate conclusions about the global methane cycle or about regional emissions estimates.

There is obviously significant overlap with the work of Sherwood et al. (2016&2021). This is not necessarily an issue as a significant new number of measurements have been reported here. However, how will the process of improving a 'definitive' source signature database evolve from here? Sherwood et al., 2017 provided a very detailed set of averages (Table 5 in their paper), however, this sort of detailed breakdown has not been made in this paper. Table 3 provides one mean value for fossil (does it also include oil?) but it is not clear how exactly this was calculated – it is an important number and surprisingly small uncertainties are reported. Table 3 is important but incomplete and needs thorough revision in relation to how averages are reported from this study (more on this in specific comments below).

The paper would benefit from some restructuring, significant clarification in places, and some further and transparent quantitative analysis.

Thank you for your feedback.
We agree that the description of the quantitive analysis could be improved, including the information on how we processed the data. In the revised version we have explained it in more detail in section 2.2.1. Section 2.2.1 was also adjusted by adding a Table that lists the variables reported in our database.

**Reviewer 1: Specific comments**

The paper would benefit from clearer structure and also clearer language in places. Section 2.2.2 might be better placed in 2.1.3? Thank you for the suggestion. We have changed the order of these paragraphs. We have also changes the titles of the paragraphs in section 3 (previously section 2.3).

Is section 2.2's 'Update of the global database' a correct section title? A 'global database' suggests a single point of data collection, however, this is not the case here? It appears that this is a separate database to the Sherwood et al. 2017/2021. Perhaps a better section title is 'Additional measurements contributing to the global isotope ratio dataset'. 2.2.1 could then be 'Structure of this database to include previous and new measurements'. Then 2.2.3 'Additional data sources from previous published literature'. These are suggestions to help clarify the work.
We agree with this suggestion and have adjusted the title of section 2.2. We have also adjusted the titles of sections 2.2.1 and 2.2.2 to more accurate statements.

Section 2.3 'results and discussion' is six lines long. Is there an error in the section structure? Section 2.4 is 'The European Methane Isotope Database' but section 2.2.1 is 'Structure of the database' Is this the same database? I think so but the understanding of the work would benefit significantly from improved paper structuring.
There was indeed a mistake in the numbering of the paragraphs, that we have corrected now. The titles have been changed to make it clearer for the reader.

Table 3 – why is Sherwood not included here? For coal Sherwood calculate a mean of -49.5 per mil published in 2017 (see their Table 5). The value in this paper in table 3 is -50.7 per mil. What has shifted the methane slightly lighter in this new database? Given the similarity in the goals of these papers it is important that the reader can make direct comparisons between them and intuitively understand the reasons for discrepancies/changes and whether

they are significant or not. How have the new measurements from this specific work shifted the global averages from Sherwood et al.? A more complete analysis of these averages and a discussion of why they have shifted would be very beneficial. Currently there is very little arithmetic behind what has actually changed between the two databases, and the significance.

Thank you for your suggestions. A more detailed comparison with the values from Sherwood et al. is now included (Table 4 and paragraph 3.2.2) and indeed it makes our study much more pertinent.

At least one of the citations in Table 3 is not relevant – I don't think Rigby et al made any conclusions on the global methane cycle, only using a value of -40 in the demonstration of the potential usefulness of isotope ratios. There is no reference to later Rigby and Turner papers here that both use 13C/12C in their global box model analyses. What about McNorton et al. 2018 etc etc? There are many other papers. It would be an interesting and very useful table if fully complete. As it stands it is highly selective of literature sources and out of date.

The original goal here was to provide a quick overview of the changes in the signatures that were assigned to the fossil fuel category, regardless of the purpose of the modelling in each study. We also wanted to show that it evolves through time, from the first global models using stable isotopes. Reacting to this referee comment we have included more references now (see Lassey et al. 2000, Neef et al. 2010, Gosh et al. 2015, McNorton et al. 2018), in an attempt to cover more of the relevant literature, even if the values that were used were the same.

Table 1 – it would be good to understand the difference between carbon and hydrogen isotopes in these quantities.

We included the numbers of hydrogen signatures in the table.

**Reviewer 1: Technical corrections**

Below are some corrections and clarifications that are needed. This is likely not comprehensive and the authors are encouraged to

improve the general readability of the manuscript.

Line 1: Rather say 'Carbon and hydrogen isotope ratio ($\delta$13C and $\delta$2H) measurements…'
This has been added.

Lines 8-9 are confusing: Previous studies are not additional literature?
We have rephrased the sentence.

Line 9: fossil fuel, not fossil fuels
This has been corrected.

Line 13: abbreviate methane as before
This has been corrected.

Line 17: Worth abbreviating GHGs?
We think it is not necessary because we use this expression only twice. It would affect the readability to introduce another abbreviation.

Line 29: 'Statistical indicators' is not a common phrase – is this correct?
We've replaced "indicators" with "methods".

Line 31: Not fully compatible? Use as simple language as possible – 'not in full agreement'?
This has been changed.

Line 36: Not a complete list. State they are selected/example publications if not citing full list.
We added "e.g." before the reference list.

Line 64: Updated version? Rather 'latest version'?
This has been changed.

Line 69 'eight'
This has been corrected.

Lines 78+: state what each of the CRDS analysers is used to measure. Were all these analysers used in a single mobile setup? Why?

Samples were taken on several mobile surveys with different and varying setups, that included one or two analysers. We have clarified this in the text (l.87-96).

Lines 104-105: Fig A1 doesn't illustrate different sampling procedures. I don't think there is any value in this figure, it is not scientific and doesn't provide useful information. Do any of the techniques used in the study need to be better explained. Are there any sampling schemes that could explain lines 78-85, for example?

We have decided to remove Fig A1, as it does not provide useful information.

Line 113 'beetween' = 'between'

This has been corrected.

Line 115 'available'.

This has been corrected.

Lines 138-140: These two sentences don't make sense

We have rephrased these sentences.

Lines 142: Literature database? This is slightly confusing. Isn't it the other way round? The literature values are incorporated into your 'European Methane Isotope Database'. This is where the use of database terms gets a bit confusing. Please clarify in the text throughout and be consistent. As I understand this the European Methane Isotope Database actually includes a global inventory of isotope signatures. Is it worth abbreviating this new database and then referring to it throughout as that?

We have changed the title to make this less confusing. We have followed your suggestion of abbreviating the term 'European Methane Isotope Database' into 'EMID', and we now refer to it as such throughout the text.

Line 148 – do you mean ' we extended the pyrogenic…'?

No, the category was "biomass burning" and we extended it to

include all pyrogenic sources. The sentence was rephrased to clarify this.

Line 154: I don't think per mille is strictly a unit. You could say 'Values are published in per mille'
This has been rephrased.

Line 156+, Section 2.2.3 – It might be good to cite in this main text all the additional literature on top of that used by Sherwood et al. or state how many additional sources there were.
We have included the number of additional literature sources in section 2.2.2, and listed them in a footnote.

Line 173: delete 'necessarily'
This has been done.

Line 190: prefer '..generally have enriched δ13C values than..'
This has been changed.

Line 208: Fig 5a (not 5A)
This has been corrected.

Line 219: 'number of measurements' not 'amount'
This has been corrected.

Lines 297+: This paragraph is confusing and it is not clear the message that is trying to be communicated. Applying appropriate weighting arithmetic is essential for what? This study provides further evidence for the values and uncertainties that are needed on source signatures if these measurements are to be used properly in to-down studies. I think that should be the main takeaway.
Applying appropriate weighting arithmetic is essential "for deriving a representative concept of the isotopic composition of CH4 sources and the associated uncertainty". We have rephrased this paragraph, the last paragraph of the conclusion and the abstract to include your suggestion as the main takeaway.

Line 318: Suggestion: 'The present database can be used in CH4 source attribution studies at local and regional scales, and to derive

global source signatures for input to global methane cycle modelling studies'

Thank you for the suggestion. We have implemented it.

Line 530: 'CH4 Sotopic' = isotopic

This has been corrected.

**Reviewer 2: General comments**

The article presents the data of methane isotopic air samples collected in Europe, their analysis, and discussions in respect to previous studies of global inventories/data. The isotopic information for various sources in methane are useful in understanding the methane budgets better. Atmospheric inverse models (top-down methods) would especially benefit from such information – not only closing the total budgets based on total methane concentration data, but also allow to separately
estimate emission magnitude of different source sectors. As authors point out, the previous studies on methane isotope measurements were mainly based on samples from US. The additional data from Europe therefore increases geographical representation, and is highly valuable. The paper presents important information and data for carbon cycle community, and is worth publishing. I also appreciate the authors for making the data open access.

I have a few suggestions below to improve the presentation of the manuscript that would possibly increase the value of the paper.

- Please consider rephrasing the title. To me, updating global inventory is a by-product, and the European data collected/presented here is the most important part.
We have changed the title to: New contributions of measurements in Europe to the global inventory of the stable isotopic composition of methane

- Please consider focusing more on Europe. You could, for example, add information and discussion about European data from previous studies, i.e. validation and updating information on Europe. It is unclear from the current manuscript how much information/data were available in Europe previously, and what were the isotopic values. You can also add comparison of the European data to global data, i.e. discussion on geographical differences/similarities of isotopic composition. In addition, geographical bias still remains within Europe. Please comment on it – what should we still improve in Europe? Where would be critical locations, and from which sources that we should sample data

from, and why?

A short description and outlook, specific to European emissions, has been added (l.239-). The analysis of the European data is developed in §3.1. Our data provide new information to better constrain specifically waste (l.210-227) and fossil fuels sources (l.229-238). Further improvements are mainly through more d2H measurements, as it is stated in the conclusion.

- Please consider focusing on new information brought by newly sampled and collected data. A few suggestions/comments on this are found under Specific comments below.

See answers to specific comments.

- Please add temporal information about the collected data in MEMO2 and those compared (collected from previous studies). If those are from very different period, the differences in isotopic composition may indicate temporal changes in underlying processes. Please comment on such if any.

We specified the sampling period in the method section (l. 85). We didn't target sources that were previously measured to study temporal changes, and if investigated in the literature, these processes were not explicitly analysed in our database (l.185).

- Modern microbial section: in many cases, CH4 emissions from natural sources (e.g. peatlands) are separately estimated from anthropogenic sources (e.g. waste, agriculture). It would be more helpful if you could analyse those separately. Could you separate those e.g. when comparing to previous studies (incl. Section 2.5)?

Modern microbial emissions are of the same formation process, and cover similar isotopic signatures. Differences between the different types of microbial emissions were mostly identified in the wetland and waste categories, and explored in more detailed in §3.2.2.

**Reviewer 2: Specific comments**

Please check language in general. Sometimes informal structures/ phrases are used. Please check the journal criteria for citations.

There are a few "in review" papers that are cited several times, but the manuscript/preprint are not available. Note that those should not be included in the final version of this manuscript. Please acknowledge that some comments maybe senseless because of this, and I apologies for that. In addition, some are cited within a page, and some at the end of the manuscript. I think all should be given at the end of the manuscript.

We cited 3 papers that were in review. Two of them are now published, and one is accepted by the journal and will be published soon. The references have been adjusted.

P1 L2: "measurements" → "isotopic measurements"
This has been revised.

P2 L44: What do you refer to by "them"?
This has been clarified.

L3 L60-61: "numerous CH4 sources could be sampled for isotopic measurements"
I think what you sample is air, and not the sources. Please revise the sentence.
The sentence has been revised.

Method:
In the excel file, I see that in some locations, only 13C or H isotopes are sampled/analysed (for MEMO2 data). Why both were not sampled at all locations? Could you add information about selection criteria or sort if there was such?
We have explained this in the beginning of §3.1 (l.193)

Section 2.1.1
- Please consider adding a table about different sampling methods, and give focus to the differences/similarities in those methods in the text. This way, you could reduce amount of details and avoid duplicates in the text.
The measurement methods are well explained and detailed in separate studies. It is not the purpose of this paper to compare the methods.
- Was there any location where different sampling methods were

applied at the same place? What kind of differences would occur in the measured values due to the differences in sampling methods?

Several methods were applied sometimes during the same campaigns, and rarely at the same locations. The comparison of the methods are in this case made for the purpose of a specific study on this campaign. For the dataset, we retained the measurements made with the IRMS, either at RHUL or UU since we have worked on bringing the systems to the same scale.

P5 L116-119:
I am now confused. In the beginning of this section (2.1.2), it is written that "the mass spectrometry measurements were performed at two laboratories". However, is it so that actually four labs analysed the air samples? You have included detailed comparison and measurement precision on IRMS, but how about CRDS compared to IRMS? Was there any differences between UHEI and LSCE measurements?

The laboratories that perform CRDS measurements had their own calibration procedure, both described in Hoheisel et al. (2019) using cylinders that were calibrated at the Max Planck Institute in Jena. It is beyond the scope of this paper to do an inter-comparison and discuss the calibration between the different labs. We did do the round-robin inter comparisons for the MEMO2 project (https://h2020-memo2.eu/wp-content/uploads/sites/198/2018/12/MEMO2-D2.1-Isotopic-measurements-linked-to-common-scale-final.pdf), which showed that both the UHEI and LSCE instruments were in the correct ball park for source measurements, and at least well-within the error bounds of each source category being isotopically characterised. it is beyond scope to include a comparison table here. We anyway report measurements from laboratories that all are on the international scale and we assume their respective calibration methods can be trusted.

P5 L133: "we did not work towards a uniform procedure"
Why? Did you allow e.g. each lab to calculate based on their choice of method? All methods being "statistically valid" does not mean that there are no differences. So I do not see why this is a valid statement to use different procedures.

Yes, we used the data that were provided by the different

laboratories, and they use slightly different approaches. That is because we don't want the data we show here to be different than in other studies from the labs who produced it. From a broader perspective, we are comparing new data with databases that include studies that are decades old and used very different measurement methodology, and without the possibility to do laboratory inter comparison of methane isotopic scales. We acknowledge that there are differences between the methods, and this problem persists in many comparative studies. The term "statistically valid" refers to the fact that all these methods minimise errors between the data and the fit, and this can be done in slightly different ways. The resulting small differences will not affect the main message. We have explained this more clearly in the text.

P6 L139: "our objective concern only values … of emitted CH4" Do you mean that the forests, where samples are taken from, also emit CH4? Most of forests are net sink of CH4, but in some cases they emit CH4. Sorry for this picky comment, but I am simply interested in. Could you provide the references where those samples were taken from?

We included 2 source signatures from aircraft measurements above a boreal forest, reported in a 2011 study by Umezawa et. al.: Umezawa, T., Aoki, S., Kim, Y., Morimoto, S., Nakazawa, T., 2011. Carbon and hydrogen stable isotopic ratios of methane emitted from wetlands and wildfires in Alaska: Aircraft observations and bonfire experiments. J. Geophys. Res. 116, D15305. https://doi.org/10.1029/2010JD015545

P6 L141-143:
I do not see country/region in Table 1. What is the "region" in those sentences?

The reference was wrong. We have now included an additional table with the description of all parameters.s

P8 L164: (2017-2020)
Is this the project period or actual period when data was collected? As in the earlier comments, it would be informative to state clearly the sampling period.

Yes, the data was collected during the years the project went on.

The sampling period is now stated in section 2.1.1 (l.85).

P8 L165-167: "The first version was made accessible on October 1st 2020"
Now that it is updated, is this information needed?
We think it helps to understand how the data was already used in other publications before this paper.

P8 L166: "The European data was used in several publications"
This gives an impression that the data has been published already. Do you mean the first version of the data was used in those publications? If so, why did you need to update? Please also consider moving this to Introduction by add a bit more details about those studies, e.g. how and what the data (e.g. which isotopologue/country) were used, and what are the main findings. This would then show the importance of the European dataset and what new information it can bring.
Parts of the data have been used in previous scientific publications, and as required these days, the data used in the publications have been made available related to these publications. In this article, we present a consolidated and completed version of all measurements that were made within our project, and additional data that we found in the literature and that were previously not included in the global database. We have moved this part in the introduction (l.66) and it indeed helps to clarity our motivation and objectives.

P9 L183-187:
What were then the isotopic signature values for those coal reservoirs where natural gas of microbial origin are present? You mention "a relatively enriched δ2H (>-250 ‰), and relatively depleted δ13C (<-60 ‰)", but are those values those measured in this study or from Milkov and Etiope (2018)?
The median values in Silesia were δ13C=-49.6 ‰ and δ2H=-180 ‰, and for Romanian gas δ13C=-57.4 ‰ and δ2H=-196 ‰. The study from Milkov and Etiope (2018) provided ranges of isotopic values to classify the formation pathways. Isotopic signatures of δ2H > -250 ‰ and δ13C < -60 ‰ indicate microbial origin via the CO2 reduction pathway. The values are discussed in comparison to western Europe later in the paragraph (p. 12, l. 231-240)

P10 L190-202:
Similar to the comment above. What is the value found in this dataset? You mention that "microbial fermentation range specified in previous reviews", but how were the waste-related source signatures compared to fermentation range measured in this study? What are the mean and rage of signature values in sewage treatment plants and biogas plants? How are those values compared to those in other types of plants?
We have specified the values in the text (p.11-12, l. 212-218). Thank you or the suggestion, it makes our argument stronger.

P10 L207-P11 L208: "This distinction is also visible in the histograms of the European Methane Isotope Database in Fig. 5.A"
- "Fig. 5.A." → Fig. 5(a)"
- From the figure, those from Poland and Romania have two peaks. Does this mean that within Poland and Romania, there are different types of plants, one similar to those in the UK/Netherlands and another that is microbial origin?
Indeed some CH4 emissions in Poland and Romania are of thermogenic origin as well, with higher d13C values. However they belong to one peak, with an average around -49 per mil. The impression of seeing 2 peaks arises  because the histogram bars are stacked, and the peak of western fossil fuel signatures is shown below, around -40 per mil.

P10 L208-209: "In western Europe, δ13C allows for a good separation between microbial and fossil fuel sources"
Is it true? I see a quite much overlap still.
Yes, the ranges of isotope signatures are well defined, and with high-precision measurements it is generally easy to distinguish microbial sources from fossil fuel based on d13C-CH4 being lower and higher than in background air, respectively. Numerous studies have successfully used d13C for source attribution in Western Europe: Zazzeri etc al. 2017, Xueref-Remy et al. 2020, Lowry et al. 2020, etc.

P11 L216-217: "increase in number of measurements"
I do not see from the figure how much data is increased, but only

the total. Is the Figure caption or text wrong?
The text was wrong. We have corrected the sentence.

Section 2.5:
Much of the text seems to be more suitable as Introduction. It is unclear what is the results of this study in addition to Sherwood et al (2021). Did representativeness increased by additional data found in this study?
The information in this paragraph does not fit in the Introduction because it updates the previous knowledge based on the new data. We rephrased parts of the paragraph to emphasize the relevance of the new database.

Section 2.6:
There are a lot of discussion where newly sampled and collected data are compared to Sherwood et al. (2017, 2021) values. However, I feel that it is difficult to comprehend the differences from the current figures and tables. Could you consider adding figures corresponding to e.g. Fig. 4 and 5, but showing e.g. differences to the previous study? Such figures could be in the supplementary, but would be helpful for readers who does not remember all the details in Sherwood et al. (2017, 2021).
Thank you for the suggestion. On Figure 4, we have added maps with the increase in amount of data from each country.

P13 L255: "mainly following the fermentation pathway"
Fermentation pathway applies to agricultural sources, but not for e.g. wetlands. Please revise the sentence.
The isotopic signature of CH4 emissions from wetlands mostly falls in the range of signatures typical of acetate fermentation, as defined in Milkov and Etiope (2018). This is the message of this sentence, and we rephrased it to make it clearer.

P13 L256: "They show a normal distribution"
Have you checked whether they really are normally distributed?
We rephrased the paragraph based on other comments and the normal distribution issue not mentioned anymore.

P13 L256-258:

I see quite much overlap in waste sector to, e.g. agriculture, also in MEMO2 data.

This is correct, we removed that statement.

P13 L271-273:

I am not sure what you wish to emphasise here. Why δ13C-CH4 from fossil fuel burning being "more variable than biomass burning" lead to "smoother" distribution?

We explained this more clearly in the revised version (p. 17, l. 332-334)

P13 L273-276 (and P13 L264-267):

I feel this relation and need of additional data on δ2H-H2O is more suitable to be mentioned in Conclusions.

Our conclusions are rather general, so we think this specific statement should stay in the Results and discussion.

P14 L280: "weighted average"

What is the weight used? If it is emission weighted, did you also measure emissions at the same locations?

The weights we used are stated in the same sentence: "the relative emission from conventional and coal fuels production worldwide (Relative weights of 0.66 for conventional fuels (oil and natural gas) and 0.34 for coal. Emission data from Saunois et al. (2020))"

P15 L297-301:

This is more suitable to be mentioned in Conclusions.

This has been changed accordingly.

Conclusions: Now that the MEMO2 project has ended (if I understood correctly), is there a plan to continue activities on isotopic measurements?

The isotopic measurements are being continued by the labs involved, but there is no follow-up project of the same scale as MEMO2.

Figures and tables

Figure 2: The literature values seem to be illustrated by boxplot-type. Are those squares mean/median? What does the bar length

illustrate? The letters presenting the types of shaded areas (e.g MF) can be bigger, and the letter "T" is better to be straight. I cannot see clearly the area "A", but is it a white space on the right side?

We specified what the squares and error bars represent in the legend. The letters have been made bigger and the abiotic area slightly darker.

Table 3:
- Please add/separate global to European means.
- What is "sem"?
- We don't think the European means are relevant here because our goal is to specifically compare the global estimates. A new table has been included in the revised version, that includes statistics for all source, for the EMID and the global database separately.
- "sem" means "standard error of the mean". We have added this in the legend.

Please consider adding A2 in the main text.

We have changed Table 2 (previously Table 1) to include the numbers for d2H. This provides information on the amounts of d13C compared to d2H data. We discuss these differences per categories rather than per country, so Figure A2 would not be well suited to directly illustrate the main text.

**Reviewer 2: Technical comments**

P2 L17: "the earth's" → "the Earth's"

This has been corrected.

P5 L115: "deliverable report publically availabe" → "deliverable report, and publically available"

This has been corrected.

---

## Referee Report (RR1)

**Review on Zheng et al., essd-2022-30**

The authors have addressed the points raised by the previous reviewers well. The significance and novelty of the manuscript is clear and the quality of the manuscript is improved. I only have some minor suggestions to be checked before publications.

L8-9: "We combined our new measurements with the last published database of CH4 isotopic signatures, as well as with additional literature, and present a new global database."
→ "We combined our new measurements with the previously published database of CH4 isotopic signatures, and present a new global database."

L55: "database can be completed" → "database can be improved"

L73-74: "We also searched the literature for more measured CH4 source signatures to add to the dataset." → "We also included the measured CH4 source signatures from other literature."

L354: "The new data" → "The new dataset"

L362: "utilizing both $\delta^{13}C$ and $\delta^{2}H$ for $CH_4$"
Do you perhaps mean "utilizing both $\delta^{13}C$ and $\delta^{2}H$ of $CH_4$ measurements", "utilizing both $\delta^{13}C$ and $\delta^{2}H$ for $CH_4$ emission estimates" or something else?

Notations: Sometimes you use "$\delta^{13}C_{CH4}$", and sometimes "$\delta^{13}C$" (similarly for $\delta^{2}H$). Unless you intended, please remember to synthesize all the notations.

Figure 5: Could you replace labels (a) and (b) to e.g. bottom/top of the panels? Now it looks as if (a) for top-left and (b) for top-right only, rather than columns. Alternatively, you could consider adding (c) and (d).

---

## Author Response (AR2)

The authors have addressed the points raised by the previous reviewers well. The significance and
novelty of the manuscript is clear and the quality of the manuscript is improved. I only have some
minor suggestions to be checked before publications.
Thank you for your feedback.

L8-9: "We combined our new measurements with the last published database of CH4 isotopic
signatures, as well as with additional literature, and present a new global database."
→ "We combined our new measurements with the previously published database of CH4 isotopic
signatures, and present a new global database."
We have partly changed the sentence following your suggestion, but we want to clearly state that additional data from the literature was also added.

L55: "database can be completed" → "database can be improved"
This was changed accordingly.

L73-74: "We also searched the literature for more measured CH4 source signatures to add to the
dataset." → "We also included the measured CH4 source signatures from other literature."
Thank you for your suggestion, the sentence reads better and was changed accordingly.

L354: "The new data" → "The new dataset"
This was changed accordingly.

L362: "utilizing both $\delta^{13}$C and $\delta^{2}$H for CH$_4$"
Do you perhaps mean "utilizing both $\delta^{13}$C and $\delta^{2}$H of CH$_4$ measurements", "utilizing both $\delta^{13}$C and
$\delta^{2}$H for CH$_4$ emission estimates" or something else?
Thank you for spotting that mistake. We meant "utilizing both $\delta^{13}$C and $\delta^{2}$H of CH$_4$"
and it was changed in the new version of the manuscript.

Notations: Sometimes you use "$\delta^{13}$C$_{CH4}$", and sometimes "$\delta^{13}$C" (similarly for $\delta^{2}$H). Unless you
intended, please remember to synthesize all the notations.
All the notations have been revised.

Figure 5: Could you replace labels (a) and (b) to e.g. bottom/top of the panels? Now it looks as if (a)
for top-left and (b) for top-right only, rather than columns. Alternatively, you could consider adding
(c) and (d).
We chose to use a, b, c and d. A new figure and caption are in the new version of the manuscript.